# The Role of International Databases in Understanding the Aetiology and Consequences of Differences/Disorders of Sex Development

**DOI:** 10.3390/ijms20184405

**Published:** 2019-09-07

**Authors:** Salma Rashid Ali, Angela Lucas-Herald, Jillian Bryce, Syed Faisal Ahmed

**Affiliations:** 1Developmental Endocrinology Research Group, Royal Hospital for Children, University of Glasgow, Glasgow G51 4TF, UK; 2Office for Rare Conditions, University of Glasgow G51 4TF, UK

**Keywords:** DSD, networks, registries

## Abstract

The International Disorders of Sex Development (I-DSD) and International Congenital Adrenal Hyperplasia registry (I-CAH) Registries were originally developed over 10 years ago and have since supported several strands of research and led to approximately 20 peer-reviewed publications. In addition to acting as an indispensable tool for monitoring clinical and patient-centered outcomes for improving clinical practice, the registries can support a wide nature of primary and secondary research and can also act as a platform for pharmacovigilance, given their ability to collect real world patient data within a secure, ethics approved virtual research environment. The challenge for the future is to ensure that the research community continues to use the registries to improve our understanding of Disorders of Sex Development (DSD).

## 1. Introduction

There are approximately 5000–8000 rare diseases or conditions that affect 6–8% of the population [Regulation EC No. 141/2000, 2000]. Patients with these conditions are often isolated and experience significant obstacles in accessing high quality healthcare and information. Research activity for these conditions is often poorly coordinated, insufficiently powered, and lacks quality and effectiveness. The group of conditions that are usually included within the broad umbrella of Disorders of Sex Development (DSD) is a typical example where there is a substantial lack of knowledge about aetiology and long term outcome, and initiatives in research are limited by the low prevalence of individual conditions. Whilst centres of expertise can bring together multidisciplinary competencies around the needs of patients, the creation of virtual networks is becoming a necessity to coordinate the work of these nodes, speed up the pace of discovery by pooling scarce resources, and ensure patient empowerment by creating a common strong voice on behalf of these patients. Data collection and archiving from a wide range of sources can bridge this knowledge gap. Often this can take the form of rare disease registries; however, registries can exist in a number of different forms. They can exist as detailed disease registries; a registry that only collects core data; a registry that simply performs e-surveillance; or a registry that links data from existing sources. This review will aim to cover how registries are helping in supporting research and clinical practice in DSD with a specific focus on the International Disorders of Sex Development (I-DSD) and International Congenital Adrenal Hyperplasia (I-CAH) Registries. 

## 2. The Development of the I-DSD Registry

The Consensus Workshop on DSD which was jointly hosted by the European Society of Paediatric Endocrinology (ESPE) and the Lawson Wilkins Pediatric Endocrine Society of North America in 2005 highlighted the need for the creation and maintenance of a database in centres of expertise [1]. At the time, such databases did exist in many regional and national centres and had provided valuable insight into many aspects of DSD, including epidemiology [2,3,4], variation of disease expression [5], initial adjustment of parents to their affected child’s condition [6], variation of investigations, and long-term outcome [7]. However, these databases and registers lacked international uniformity, a key feature particularly desirable when dealing with a rare group of conditions. With the initial help of ESPE followed by the European Union, through an EUFP7 funded project, EuroDSD a European web-based registry and research environment for DSD was developed between 2008 and 2011. During this period, the registry gained international popularity beyond Europe, and from 2011 to 2017 it received funding from the Medical Research Council (MRC) to become the I-DSD Registry. The day to day management of the I-DSD and I-CAH registries is undertaken by the Project Management Group which is guided by the joint Steering Committee which is responsible for the overall direction of the initiative. New research proposals are handled by the Project Management Group and reviewed by a Scientific Panel. Members of the Steering Committee and the Scientific Panel have fixed terms. Increasingly, the Steering Committee consists of representatives of international societies that have a dedicated DSD working group. Other members of the Steering Committee and Scientific Panel are chosen from those who are actively involved in I-DSD or I-CAH.

## 3. Description of the Cases in the I-DSD Registry

As of June 2019, there were 2456 cases with DSD registered on the I-DSD Registry. These cases are from 80 centres in 34 countries across 5 continents. Overall, 1391 (57%) are male and the highest number of cases are registered as having a disorder of gonadal development (*n* = 758, 31%) followed by disorders of androgen action (*n* = 589, 24%). Of those reported as males, the most common registered conditions are disorders of gonadal development (*n* = 384, 28%) followed by non-specific XY DSD (*n* = 337, 24%) and disorders of androgen action (*n* = 236, 17%). Of those reported as females, the most common registered conditions are disorders of androgen action (*n* = 374, 35%) followed by disorders of gonadal development (*n* = 353, 33%) and disorders of androgen synthesis (*n* = 187, 18%). The majority of the registered cases are 46, XY (*n* = 1741, 71%). Currently, the age distribution of registered cases ranges from <1 year to 77 years (median 22 years), with the highest proportion currently being over the age of 16 years (*n* = 787, 32%). Overall, comparison of the case mix that exists in the I-DSD Registry [8] to the range of cases that may exist in a specialist service [9] has shown significant similarities. 

## 4. Past and Current Research Activities of the I-DSD Registry

To date collaborations through the I-DSD Registry have generated almost 20 published articles, which have cumulatively been cited over 200 times. Each collaboration has offered new insights into the field of DSD, changing the way that clinicians approach patients affected by these conditions. One of the first published outputs from the I-DSD Registry was by Koleskinska et al. in 2014 [10]. This research not only demonstrated that the I-DSD Registry could be effectively used to pool data on DSD but it was also seminal in highlighting that the practice of assigning female sex to 46, XY infants had reduced over time and that this was irrespective of the degree of masculinisation of the external genitalia [10]. Later, Cox et al. (Cox et al. 2014) identified that as many as 27% of individuals affected by DSD may have an associated condition and that the frequency of additional associated conditions was highest in disorders of gonadal development and in nonspecific 46, XY DSD [8]. One of the commonest associated conditions was small for gestational age (SGA), which was confirmed by Poyrazoglu who subsequently reported that as many as 18% of 46, XY individuals may be born SGA [11]. These investigators also highlighted that the presence of SGA in those with XY DSD provided helpful insights into the underlying aetiology. Each of these studies highlights the increasing clinical knowledge that can be obtained via research activities using the I-DSD Registry. The importance of identifying the underlying aetiology was further reinforced in the study by Lucas-Herald et al. which showed that a molecular confirmation of a defect in the androgen receptor was critical in predicting the long-term outcome in clinically diagnosed cases of partial androgen insensitivity syndrome [12]. Collecting long-term outcome data in PAIS has been difficult to date and the utility of the I-DSD Registry was clearly proven in this context. Another challenging area that the I-DSD Registry has tackled is the long-term outcome of the gonads in cases of androgen insensitivity [13] as well as those cases with 45X/XY gonadal dysgenesis [14]. All of these research projects answer questions on DSD, which would not be possible in these rare diseases with only small local datasets. In addition, their outputs can be used to inform clinical care and health service provision for affected individuals. However, research activities from the I-DSD Registry are not just confined to studies of long-term outcome. In 2016, Hornig et al. were able to perform genetic analysis of cultured genital fibroblasts from 169 male individuals with DSD, through identification of cases in the Registry and collaboration with the local centres, where consent for analysis was given [15]. As a result, they reported a new class of individuals who were Androgen Receptor (AR) mutation negative but had a PAIS phenotype, who could be reliably identified using transcriptional induction of an AR target protein, apolipoprotein D (APOD) [15]. The Registry has also been used to identify specialists in the field of DSD through international surveys regarding current models of practice in DSD, in terms of medical and surgical management [16] as well as psychosocial care [17]. Given much of the work produced from the I-DSD Registry relies on the correct input of data by Registry users, confirmation of the quality of the data is of course essential [18]. As such, the quality of the data in the Registry has recently been reviewed, demonstrating that it has a high degree of accuracy [19] and thereby confirming the validity of the information included in associated studies. There are currently 9 ongoing studies that are using the I-DSD Registry, with the number of studies starting per year increasing annually (from 1 to 5 per year on average). Further details of these studies are available on the I-DSD website (https://home.i-dsd.org).

## 5. The Development of the I-CAH Registry 

By 2013, it was clear that the I-DSD Registry was also being used by several centres to recruit cases of CAH. However, to collect information on CAH, the fields within the I-DSD Registry were not completely suitable and, in addition, given that some cases of CAH could not be classed as DSD, an international network of researchers and clinicians looking after people with congenital adrenal hyperplasia (CAH) was created to facilitate the development of a CAH module within the I-DSD Registry. In addition, the I-CAH Registry could be entered through a separate website, entitled the *I-CAH Registry*. This approach allowed the registry project to match the pace of discovery of new management strategies and therapies for patients with CAH. This development continued to be managed and administered by the I-DSD governing body and was supported by the TAIN (Treatment of Adrenal Insufficiency in Neonates) project which was an EU-funded consortium dedicated to developing a new formulation of hydrocortisone. The I-CAH Registry has developed a major focus on pharmacovigilance and has also been exploring how it can be used to develop clinical benchmarks. The longitudinal module in the I-CAH registry has also often been used as a clinical tool in a clinic setting, highlighting the power of disease registries in guiding clinical users towards standardised data collection within the routine clinical setting whilst acting as an indispensable tool within the clinical setting.

## 6. Description of the Cases in the I-CAH Registry

As of June 2019, there were 1429 cases of CAH registered in the I-CAH Registry. These cases are from 65 centres in 29 countries across 5 continents. Overall, 815 patients (57%) are female. A karyotype of 46, XX was registered in 756 (53%) of cases. The majority of registered cases are attributed to 21-hydroxylase deficiency (*CYP21A2*) (*n* = 1312, 92%), followed by 11β- hydroxylase deficiency (*CYP11B1*) (*n* = 55, 4%). Other rare types of CAH account for less than 1% of all registered cases and include 3β-hydroxysteroid dehydrogenase deficiency (*HSD3B2*) (*n* = 18, 1%), 17α-hydroxylase deficiency (*CYP17A1*) (*n* = 9, 0.6%), cytochrome P450 oxidoreductase deficiency (*POR*) (*n* = 4, 0.3%), and steroid acute regulatory protein deficiency (*StAR*) (*n* = 2, 0.1%). The majority of cases within the I-CAH registry presented aged less than 3 months (*n* = 913, 64%). Currently, the age distribution of cases ranges from birth to 82 years (median 15 years), with almost half of all cases aged 16 years or over (*n* = 685, 48%).

## 7. Past and Current Research Activities of the I-CAH Registry

The development of the I-CAH Registry in 2014 has enabled pooling of data for research and multi-centre collaborations in the field of CAH, providing further insight into the diagnosis, management, and outcomes in this rare condition. An initial study evaluating the variation in practice for reaching a diagnosis of CAH demonstrated user acceptability of the I-CAH Registry, with a 10-fold increase in the number of cases registered and a temporal shift in diagnostic practice towards the use of molecular genetics [20]. The quality of the data within the I-CAH registry has also been examined and has demonstrated a high degree of validity, consistency and accuracy for conditions such as CAH, with the highest degree of internal validity for CAH compared with other registered conditions affecting sex development [19].

The longitudinal module was developed in 2015 and studies incorporating data from this module have provided a greater understanding of current medical management in patients with CAH, providing an opportunity to compare treatment regimens and outcomes amongst different centres. Physiological replacement of glucocorticoid and mineralocorticoid is vital for optimal control in patients with CAH. A lack of evidence based guidelines for salt and mineralocorticoid replacement in young children with CAH was reflected in a recent I-CAH study which demonstrated substantial variation in salt supplementation regimens, including timing of treatment, frequency and dosing, in children under the age of three years, with variations dependant on the treatment centre [21]. Similarly, data from a cohort of 269 patients from 22 centres in 14 countries confirmed variation in hormonal replacement regimens across paediatric and adult patients with glucocorticoid doses above the recommended thresholds in some age groups [22]. The management of CAH requires a balance between excess glucocorticoid leading to adverse effects and a deficiency of glucocorticoid leading to a risk of adrenal crises. The I-CAH longitudinal module can also be used to collect prospective data on the occurrence on adverse events in CAH, including sick day episodes and adrenal crises. A preliminary analysis of this data has demonstrated that patients with CAH experience frequent sick day episodes [23] and further studies are ongoing to determine factors which may be associated with the occurrence of adrenal insufficiency related adverse events. There are currently seven ongoing studies that are using the I-CAH Registry including industry supported-studies that are examining the feasibility of studies using novel therapeutic options in CAH. Further information on future studies and developments can be obtained from the I-CAH website (http://home.i-cah.org).

## 8. The Respective Role of Detailed Disease Registries, Core Registries and Surveillance Systems for Studying DSD

DSD are a group of relatively rare conditions with a wide spectrum of pathophysiology. These conditions pose a challenge due to gaps in knowledge of long-term outcomes and a lack of evidence based multidisciplinary care. Rare disease registries enable pooling of data for research, healthcare surveillance, multicentre collaboration, and the development of best practice guidelines. Participation in international registries for rare conditions such as DSD is encouraged [24,25] and has been endorsed by many European and international initiatives such as EURODIS (European Organisation for Rare Diseases), EUCERD (European Union Committee of Experts on Rare Diseases) and IRDiRC (International Rare Diseases Research Consortium). A recent survey amongst reference centre leads of the European Reference Network for Rare Endocrine Conditions (Endo-ERN) sought to evaluate the level of engagement in local, national, and international disease registries for rare endocrine conditions [26]. This exercise highlighted that despite a high level of awareness of registries for conditions such as DSD within these expert centres, participation in registries was relatively low. There may be several factors contributing to the low participation rates including lack of patient consent, lack of sufficient time to recruit patients or enter data [16] or the absence of transparent quality indicators for the registries [27]. Optimising participation in registries while encouraging record completion and data entry at regular intervals may be achieved by improving the registry user interface, reducing the number of fields required for completion, or perhaps focusing on completion of core information. Furthermore, consideration should be given to methods of sustaining registries over the longer-term including membership fees, dissemination of information through newsletters and increased stakeholder/ patient involvement. The biennial international I-DSD meeting has also been very helpful in providing the I-DSD and I-CAH users with a community spirit and a common vision.

Based on the lessons learnt in I-DSD, and with support from Endo-ERN, the European Society for Paediatric Endocrinology and European Society of Endocrinology, the endocrine community in Europe has embarked on a new project funded by the European Union’s Health Programme, EuRRECa (European Registries for Rare Endocrine Conditions, eurreca.net). The aim of this project is to maximise the opportunity for patients, health care professionals, and researchers to participate in high-quality, patient-centred registries for DSD and other rare endocrine conditions covered within Endo-ERN. The two main arms of the projects are: 1. A core endocrine registry that collects a core dataset based on standardised data entry with signposting to high quality, detailed, disease-specific registries, 2. An e-reporting programme for rare endocrine conditions (e-REC) which enables measurement of core indicators of ERN activity. Thus, in addition to rare disease registries such as the I-DSD and I-CAH Registry, there may be a place for a core registry that only collects core patient information (e.g., EuRRECa Core Endocrine Registry) and which directs people to detailed disease registries and which is supported by an even more light-touch surveillance system that enables e-reporting of conditions. This pathway of linking registries within a defined network will allow the possibility of long-term sustainability and improve the quality and validity of registries with the overall aim of optimising patient care and research. Furthermore, linking existing registries would provide current platforms with options to select which data are shared. Registries would only be able to be linked if they had common data elements, thus the linking of data would also lead to greater standardisation of data and its quality. Linking of registries would also lead to greater clarity and transparency of data access as it is likely that only registries with stringent governance protocols will be in a position to share data. 

## 9. Examples of How Studying Rare Conditions Such as DSD & CAH Can be Performed at A Local, National, and International Level Through a Wide Range of Platforms

Knowledge of conditions such as DSD and CAH can be gleaned through a wide range of databases. Studies performed in the early 2000s using linked data sets within the National Health Service in Scotland allowed the development of a congenital anomaly register that was designed specifically to collect information on genital anomalies. These studies provided helpful epidemiological data and supported the development of a national clinical network in Scotland [2]. Whilst epidemiological studies have a clearly important role, their role is limited when it comes to understanding variations in management and outcome for a heterogeneous group of rare conditions within a relatively small population. In Scotland, the formation of a clinical network led to the development of a local disease registry which was used to understand clinical practice nationwide [6,28] and this registry was superseded by the I-DSD Registry. At a regional level, the Scottish DSD network has focussed more over the last 5 years on performing an electronic surveillance exercise within Scotland monitoring the presentation of newborns with atypical genitalia [29]. This exercise has allowed it to combine an epidemiological objective with an objective related to service improvement. The relatively small population of Scotland combined with the existence of defined clinical networks has facilitated this surveillance exercise.

On the other hand, detailed meaningful studies of a condition such as Congenital Adrenal Hyperplasia (CAH) which has an estimated prevalence of 1 in 15,000 births [30] would not be possible in Scotland, which has an annual birth rate of about 55,000. The majority of cases are inherited in an autosomal recessive manner, with more than 90% of cases occurring secondary to 21-hydroxylase enzyme deficiency [31]. Clinical features of classical CAH include androgen excess with resultant genital ambiguity in females, precocious puberty, infertility and reduced quality of life in affected individuals [32]. Furthermore, in CAH, the resultant adrenocortical enzyme deficiency provides patients with a life-long risk of adverse events including sick day episodes and ensuing adrenal crises [33], the leading cause of mortality in CAH [32]. There is little evidence on the occurrence of adverse events (sick day episodes and adrenal crises) in children with CAH from large multi-centre cohort studies. Data on the frequency of adverse events in adrenal insufficiency have been collected in different settings such as local centres [33] as well as at a national level [34]. The I-DSD and I-CAH Registries provide an ideal platform for the study of core clinical outcomes at a global level, with the overall aim of better informing the clinical and patient community and laying a foundation for which future studies can target interventions aimed at improving the quality of life of children with CAH.

## 10. Example of How Disease Registries Can Develop a Network that Provides Other Benefits 

Disease registries, in particular those integrated at international level, are pivotal in the development of networks as centres of expertise. This is particularly important in the field of rare conditions as these resources facilitate the pooling of specialists around the world as well as concentrating data in patients with rare conditions. I-DSD is an example of such a registry which has developed a network of experts that collaborate across borders to discuss expert management of similar cases. Registered users and the wider community have regular (biennial) symposia that are held around the world, with the 2019 meeting being held outside Europe for the first time (Sau Paulo, July 2019). The I-DSD Registry also facilitated the development of a network of DSD centres and specialists that formed the backbone of initiatives such as the I-DSD network, COST DSDnet and the reference centres within the European Reference Network for Rare Endocrine Conditions (Endo-ERN) and supported a fixed term project, DSDlife, that studied the general quality of life and psychological well-being as well as the quality and satisfaction with past treatments in adult patients with DSD (www.dsd-life.eu). I-DSD and COST DSDnet also combined forces to organize a patient workshop which explored several issues including the need for research and the use of registries [35]. In addition to DSDlife and DSDnet, the I-DSD Registry also complemented other EU projects such as EU-TAIN (Treatment of Adrenal Insufficiency in Neonates) which led to the development of I-CAH, an international registry for CAH.

## 11. Future Direction-Quality, FAIR Values, Inclusion and Sustainability 

Whilst over a decade ago, there was relatively little interest in detailed disease registries for rare conditions, there are now over 800 rare disease registries in Europe alone. The emphasis on registries has been further strengthened with the inclusion of participation in rare disease registries as a quality indicator of a reference centre. The utility and long-term sustainability of a detailed disease registry can increase significantly if the data that are stored within the registry adhere to the principles of FAIR data (findable, accessible, interoperable and reusable). This involves designing it to comply with international standards of quality, structure and content, access control, and also adopting common methods and processes for information/patient discoverability, sharing and federation with other registries. Thereby, users can more easily compare, pool, and analyse patient datasets, using sufficient numbers of cases for meaningful clinical research and public health purposes in areas identified as high priority by the DSD community [36]. Going forward, it will become particularly important for the I-DSD and I-CAH registries to maximise their quality [18] and increase the participation of patients and professionals in activities that are aimed at improving clinical care as well as our knowledge of these conditions A high quality disease registry also requires clear governance structures and scrutiny and the effort involved in providing this is often under-estimated. The joint management of the I-CAH and I-DSD registries as well as the sharing of a common platform has maximised efficiency and effectiveness whilst ensuring that patients, clinicians and researchers continue to remain fully involved in their respective areas of interest. This strategy can be applied to other conditions that may affect sexual development and are not traditionally considered as DSD.

## 12. Summary

In summary, the I-DSD and I-CAH registries are powerful tools for improving our knowledge of a group of rare conditions where research, particularly in the field of outcome, has been hampered by the inherent rarity and heterogeneity of the conditions. The registries have also led to the development of a strong clinical and research network and it is anticipated that the wider user community will continue to use the registries to develop new studies within all related disciplines.

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
