# Peer review of "The Role of International Databases in Understanding the Aetiology and Consequences of Differences/Disorders of Sex Development"

_ijms, 2019, doi:10.3390/ijms20184405_

Round 1

Reviewer 1 Report

Comments on the manuscript: The Role Of International Databases In  Understanding The Aetiology & Consequences Of  Differences/Disorders of Sex Development

The manuscript is a description of the history of the I-DSD and I-CAH Registries, which could be potentially useful for the spread of the knowledge regarding DSD in humans.
the manuscript is well written and includes a lot of information regarding the use and the evolution of such databases. They also provide examples of collaborations generated by those e-bases.

My concern is that there is no new knowledge generated in this manuscript, only the advertising of the benefits of electronic disease databases, collaborations among groups and countries.   The authors also included data regarding the data existent in the database, which is public and can be easily obtained by the people interested.

However, I´m not sure that a scientific journal is the right place for this kind of manuscripts. In fact, I think that I would say that it is not. It is not scientifically novel and also no provides new tools or procedures to be employed in the detection and treatment of DSD

Therefore, my concerns are only on the topic and not in the quality of the manuscript that it is solid to my point of view. 

Author Response

Thank you for your feedback. This was an invited review and the topic of the review had already been decided at the outset.

Reviewer 2 Report

I think this is an important paper which describes the development of international registries for DSD and CAH. Such descriptions can illuminate the use of such databases as well as serve as an inspiration and template for database development of other rare conditions. As such I have one major comment:

More detail regarding the creation of the database, governance (mentioned in line 266) and the different approaches to data collection would be useful. In line with the mechanics of a registry, the authors allude to poor participation in the registries, and a small description regarding lessons learned: what works/what does not work in creating registries and optimizing participation would be useful

A minor comment: Line 56 and line 60 mention the participants as male or female. It would be useful to know how they defined male/female and whether there were any non-binary participants. 

Author Response

Reviewer 2

I think this is an important paper which describes the development of international registries for DSD and CAH. Such descriptions can illuminate the use of such databases as well as serve as an inspiration and template for database development of other rare conditions. As such I have one major comment:

More detail regarding the creation of the database, governance (mentioned in line 266) and the different approaches to data collection would be useful. In line with the mechanics of a registry, the authors allude to poor participation in the registries, and a small description regarding lessons learned: what works/what does not work in creating registries and optimizing participation would be useful

Response

(Added on Page 2, Lines 53-62 within the section ‘The development of the I-DSD Registry’)

The day to day management of the I-DSD and I-CAH registries is undertaken by the Project Management Group which is guided by the joint Steering Committee which is responsible for the overall direction of the initiative. New research proposals are handled by the Project Management Group and reviewed by a Scientific Panel. Members of the Steering Committee and the Scientific Panel have fixed terms. Increasingly, the Steering Committee consists of representatives of international societies that have a dedicated DSD working group. Other members of the Steering Committee and Scientific Panel are chosen from those who are actively involved in I-DSD or I-CAH.

Response

(Added on Page 4, Lines 193-199 within the section ‘The respective role of international detailed disease registries, core registries and surveillance systems for studying DSD’)

Optimising participation in registries, encouraging record completion and data entry at regular intervals may be achieved by improving the registry user interface, reducing the number of fields required for completion or perhaps focusing on completion of core information. Furthermore, consideration should be given to methods of sustaining registries over the longer-term including membership fees, dissemination of information through newsletters and increased stakeholder/ patient involvement. The biennial international I-DSD meeting has also been very helpful in providing the I-DSD and I-CAH users a community spirit and a common vision.

A minor comment: Line 56 and line 60 mention the participants as male or female. It would be useful to know how they defined male/female and whether there were any non-binary participants. 

Response

(Text edited on Page 2, Lines 67 and 69)

Information regarding participant gender was extracted directly from the fields within the I-DSD and I-CAH Registries that are completed by the clinician at the time of case entry; options are Male/Female/Unknown. Currently, there is no option to select ‘non-binary’ amongst the responses. In the revised version of the I-DSD and I-CAH registries there will more options including ‘Non-binary’, ‘Other’, ‘Asked but unknown’. However, the text on Page 2, Lines 64 and 66 has been altered slightly.

Reviewer 3 Report

This review by Ali et al. on the role of international databases is well written and concise. It summarizes the intention, advantages and achievements regarding the development of the I DSD and I CAH registries.

Please find a very few minor suggestions

-regarding the content of the manuscript:

your describe the positive aspect of registries, but you only touch superficially the problems in lines 181 ff.

For the sake of comprehensiveness, please add a more detailed description of the obstacles and hurdles for physicians regarding participation in such databases and add a suggestion how these problems could be solved.

The idea of linking existing national registries sounds perfect. Nevertheless, there may be reluctance in those countries who already have functioning databases. It would be interesting to know, what are the arguments for those functioning databases to merge with international ones.

-regarding formal aspects of the manuscript:

There is some redundancy in lines 217-233 regarding CAH. This passage could be shortened.

Author Response

Reviewer 3

This review by Ali et al. on the role of international databases is well written and concise. It summarizes the intention, advantages and achievements regarding the development of the I DSD and I CAH registries.

Please find a very few minor suggestions

-regarding the content of the manuscript:

your describe the positive aspect of registries, but you only touch superficially the problems in lines 181 ff.

For the sake of comprehensiveness, please add a more detailed description of the obstacles and hurdles for physicians regarding participation in such databases and add a suggestion how these problems could be solved.

Response

(Added on Page 4, Lines 193-199 within the section ‘The respective role of international detailed disease registries, core registries and surveillance systems for studying DSD’)

Optimising participation in registries, encouraging record completion and data entry at regular intervals may be achieved by improving the registry user interface, reducing the number of fields required for completion or perhaps focusing on completion of core information. Furthermore, consideration should be given to methods of sustaining registries over the longer-term including membership fees, dissemination of information through newsletters and increased stakeholder/ patient involvement. The biennial international I-DSD meeting has also been very helpful in providing the I-DSD and I-CAH users a community spirit and a common vision.

The idea of linking existing national registries sounds perfect. Nevertheless, there may be reluctance in those countries who already have functioning databases. It would be interesting to know, what are the arguments for those functioning databases to merge with international ones. 

Response

(Added to Page 5 Lines 215-220 in the Section on ‘The respective role of international detailed disease registries, core registries and surveillance systems for studying DSD’)

Furthermore, linking existing registries would provide current platforms with options to select which data are shared. Registries would only be able to be linked if they had common data elements, thus the linking of data would also lead to greater standardisation of data and its quality. Linking of registries would also lead to greater clarity and transparency of data access as it is likely that only registries with stringent governance protocols will be in a position to share data.

-regarding formal aspects of the manuscript:

There is some redundancy in lines 217-233 regarding CAH. This passage could be shortened.

Response

(Deleted in Page 6, Lines 2525-253 in the Section on ‘Examples of how studying rare conditions such as DSD & CAH can be performed at a local, national and international level through a wide range of platforms’)

Sentence removed- To address this wide range of issues for a rare group of conditions and to learn from each other’s experiences, there is a need for a global platform.

Round 2

Reviewer 1 Report

Nothing to comment